# IS SYNTHETIC DATA FROM GENERATIVE MODELS READY FOR IMAGE RECOGNITION?

**Ruifei He**[1*]  **Shuyang Sun**[2]  **Xin Yu**[1]  **Chuhui Xue**[3]  **Wenqing Zhang**[3]  **Philip Torr**[2]
**Song Bai**[3†]  **Xiaojuan Qi**[1†]
[1]The University of Hong Kong  [2]University of Oxford  [3]ByteDance

## ABSTRACT

Recent text-to-image generation models have shown promising results in generating high-fidelity photo-realistic images. Though the results are astonishing to human eyes, how applicable these generated images are for recognition tasks remains under-explored. In this work, we extensively study whether and how synthetic images generated from state-of-the-art text-to-image generation models can be used for image recognition tasks, and focus on two perspectives: synthetic data for improving classification models in data-scarce settings (*i.e.* zero-shot and few-shot), and synthetic data for large-scale model pre-training for transfer learning. We showcase the powerfulness and shortcomings of synthetic data from existing generative models, and propose strategies for better applying synthetic data for recognition tasks. Code: https://github.com/CVMI-Lab/SyntheticData.

## 1 INTRODUCTION

Over the past decade, deep learning powered by large-scale annotated data has revolutionized the field of image recognition. However, it is costly and time-consuming to manually collect a large-scale labeled dataset, and recent concerns about data privacy and usage rights further hinder this process. In parallel, generative models that aim to model real-data distributions can now produce high-fidelity photo-realistic images. In particular, recent text-to-image generation models (Nichol et al., 2021; Ramesh et al., 2022; Saharia et al., 2022b) have made major breakthroughs in synthesizing high-quality images from text descriptions. This promotes us to ask: is synthetic data from generative models ready for image recognition tasks?

There are a few early attempts at exploring synthetic data from generative models for image recognition tasks. Besnier et al. (2020) use a class-conditional GAN (BigGAN (Brock et al., 2018) trained for ImageNet-1000 classes) to generate images for training image classifiers. Zhang et al. (2021) leverage StyleGAN (Karras et al., 2019) to produce synthetic labeled data for object-part segmentation. Jahanian et al. (2021) manipulate the latent space of a GAN model to produce multi-view images for contrastive learning. Albeit promising, early works either address tasks on a small scale or only for a specific setting. Plus, they all focus on GAN-based models and none explore the revolutionary text-to-image generation models, which hold more promises to benefit recognition tasks.

In this paper, we present the first study on the state-of-the-art text-to-image generation models for image recognition. With the power of text-to-image generation, we could hopefully not only generate massive high-quality labeled data, but also achieve domain customization by generating synthetic data targeted for a specific label space, *i.e.* the label space of a downstream task. Our study is carried out on one open-sourced text-to-image generation model, GLIDE (Nichol et al., 2021) [1]. We attempt to uncover the benefits and pitfalls of synthetic data for image recognition through the lens of investigating the following two questions: 1) is synthetic data from generative models ready for improving classification models? 2) whether synthetic data can be a feasible source for transfer learning (*i.e.* model pre-training)? It is worth noting that for 1), we only studied the zero-shot and few-shot settings because the positive impact of synthetic data diminishes as more shots are present. And, we build most of our investigations on the state-of-the-art method CLIP (Radford et al., 2021) with the feature extractor initialized with large-scale pre-trained weights frozen.

---

[*] Part of the work is done during an internship at ByteDance. Email: ruifeihe@eee.hku.hk
[†] Corresponding authors: songbai.site@gmail.com, xjqi@eee.hku.hk
[1]At the beginning of this project, GLIDE is the only open-sourced text-to-image synthesis model that also delivers high-quality synthesis results.

**Our Findings.** First, in the zero-shot setting, *i.e.* no real-world data are available, we demonstrate that synthetic data can significantly improve classification results on 17 diverse datasets: the performance is increased by $4.31\%$ in top-1 accuracy on average, and even improved by as much as $17.86\%$ on the EuroSAT dataset. To better leverage synthetic data in this setting, we also investigate useful strategies to increase data diversity, reduce data noise, and enhance data reliability. This is achieved by designing diversified text prompts and measuring the correlation of text and synthesized data with CLIP features.

Second, in the few-shot setting, *i.e.* a few real images are available, albeit not as significant as in the zero-shot task, synthetic data are also shown to be beneficial and help us achieve a new state of the art. Our observation shows that the domain gap between synthetic data and downstream task data is one challenge on further improving the effectiveness of synthetic data on classifier learning. Fortunately, in this setting, the accessibility of real data samples can provide useful information about the data distribution of the downstream task. We thus propose to use real images as guidance in the generation process to reduce domain gaps and improve effectiveness.

Third, in large-scale model pre-training for transfer learning, our study shows that synthetic data are suitable and effective for model pre-training, delivering superior transfer learning performance and even outperforming ImageNet pre-training. Especially, synthetic data work surprisingly well in unsupervised model pre-training, and favor ViT-based backbones. We also demonstrate that by increasing the label space (*i.e.* text prompts) for data generation, the enlarged data amount and diversity could further bring performance boosts. Besides, synthetic data can work collaboratively with real data (*i.e.* ImageNet) where we obtain improved performance when the model is initialized with ImageNet pre-trained weights.

## 2    RELATED WORKS

**Synthetic Data for Image Recognition.** There are mainly two forms of synthetic data for image recognition, *i.e.* 1) synthetic datasets generated from a traditional simulation pipeline; 2) synthetic images output from generative models.

The first type, synthetic datasets (Dosovitskiy et al., 2015; Peng et al., 2017; Richter et al., 2016), are usually generated from a traditional pipeline with a specific data source, *e.g.* synthetic 2D renderings of 3D models or scenes from graphics engines. However, this traditional way of generating synthetic datasets has several drawbacks: 1) manually defined pipeline generated synthetic data may have a certain gap with real-world data; 2) taking up huge physical space to store and huge cost to share and transfer; 3) data amount and diversity bounded by the specific data source.

Compared with synthetic datasets, generative models are a more efficient means of synthetic data representation, exhibiting favorable advantages: 1) could produce high-fidelity photorealistic images closer to real data since they are trained on real-world data; 2) highly condensed compared to synthetic data itself, and take up much reduced storage space; 3) potentially unlimited synthetic data size. Only recently, few works attempt to explore synthetic data generated from generative models for image recognition. Besnier et al. (2020) use a class-conditional GAN to train classifiers of the same classes. Zhang et al. (2021) leverage the latent code of StyleGAN (Karras et al., 2019) to produce labels for object part segmentation. While they achieve promising results, both works are task-wise and only employed on a small scale. Jahanian et al. (2021) use a GAN-based generator to generate multiple views to conduct unsupervised contrastive representation learning. These works, however, explore upon the traditional GAN-based models; in contrast, our work investigates with the best released text-to-image generation model, which demonstrates new customization ability for different downstream label space.

**Text-to-Image Diffusion Models.** Diffusion models (Sohl-Dickstein et al., 2015; Ho et al., 2020; Nichol & Dhariwal, 2021) have recently emerged as a class of promising and powerful generative models. As a likelihood-based model, the diffusion model matches the underlying data distribution $q(x_0)$ by learning to reverse a noising process, and thus novel images can be sampled from a prior Gaussian distribution via the learned reverse path. Because of the high sample quality, good mode coverage and promising training stability, diffusion models are quickly becoming a new trend in both unconditional (Ho et al., 2020; Nichol & Dhariwal, 2021; Ho et al., 2022) and conditional (Dhariwal & Nichol, 2021; Rombach et al., 2022; Lugmayr et al., 2022; Saharia et al., 2022a; Meng et al., 2021; Saharia et al., 2022c) image synthesis fields.

In particular, text-to-image generation can be treated as a conditional image generation task that requires the sampled image to match the given natural language description. Based upon the formulation of the diffusion model, several text-to-image models such as Stable diffusion (Rombach et al., 2022), DALL-E2 (Ramesh et al., 2022), Imagen (Saharia et al., 2022b) and GLIDE (Nichol et al., 2021) deliver unprecedented synthesis quality, largely facilitating the development of the AI-for-Art community. Despite achieving astonishing perceptual results, their potential utilization for high-level tasks is yet under-explored. In this paper, we utilize the state-of-the-art model GLIDE and showcase its powerfulness and shortcomings for synthesizing data for recognition tasks.

## 3  IS SYNTHETIC DATA READY FOR IMAGE RECOGNITION?

In the following sections, we answer the question by studying whether synthetic data can benefit recognition tasks and how to better leverage synthetic data to address different tasks. We carry out our exploration through the lens of two basic settings with three tasks: synthetic data for improving classification models in the data-scarce setting (*i.e.* zero-shot and few-shot) (see Sec. 3.1 and Sec. 3.2) and synthetic data for model pre-training for transfer learning (see Sec. 3.3).

**Model Setup for Data-scarce (*i.e.* Zero-shot and Few-shot) Image Classification.** As CLIP (Radford et al., 2021) is the state-of-the-art approach for zero-shot learning, we conduct our study for zero-shot and few-shot settings upon pre-trained CLIP models, aiming to better understand synthetic data upon strong baselines. There have been a few attempts on better tuning pre-trained CLIP for data-scarce image classification, such as CoOp (Zhou et al., 2022b), CLIP Adapter (Gao et al., 2021), and Tip Adapter (Zhang et al., 2022), where the image encoder is frozen for better preserving the pre-trained feature space. We argue that different tuning methods could all be regarded as different ways of learning classifier weights, *e.g.* CoOp optimizes learnable prompts for better learning classifiers.

Here, we adopt a simple tuning method, Classifier Tuning (**CT**), a baseline method introduced in Wortsman et al. (2022). Concretely, for a k-way classification, we input the class names $C = \{c_1, ..., c_k\}$ with prompt $s_i = $ "a photo of a $\{c_i\}$" into the text encoder $h$ of CLIP to obtain the text features $h(s_i)$. Then the text features $h(s_i)$ could be used to construct classifier weights $W \in R^{d \times k}$, where $d$ is the dimension of text features. Finally, we combine the image encoder $g$ with the classifier weights $W$ to obtain a classification model $f(x) = g(x)^T W$. We empirically show that **CT** performs comparably with other tuning methods. Compared with complex designed tuning methods, we hope to use a simpler method for better investigating the effectiveness of synthetic data.

### 3.1  IS SYNTHETIC DATA READY FOR ZERO-SHOT IMAGE RECOGNITION?

Our aim is to investigate to what degree synthetic data are beneficial to zero-shot tasks and how to better leverage synthetic data for zero-shot learning.

**Zero-shot Image Recognition.** We study the inductive zero-shot learning setting where no real training images of the target categories are available. CLIP models are pre-trained with large-scale image-caption pairs, and the similarities between paired image features (from an image-encoder $g$) and text features (from a text-encoder $h$) are maximized during pre-training. The pre-trained feature extractor can then be used to solve zero-shot tasks where given an image, its features from $g$ are compared with text features of different classes from $h$ and the image is further assigned to the class that has the largest similarity in the CLIP text-image feature space.

**Synthetic Data for Zero-shot Image Recognition.** Though CLIP models exhibit strong zero-shot performance thanks to the large-scale vision-language dataset for pre-training, there are still several shortcomings when the model is deployed for a downstream zero-shot classification task, which may be attributed to unavoidable data noise in CLIP's pre-training data or the label space mismatch between pre-training and the zero-shot task. Hence, with a given label space for a zero-shot task, we study whether synthetic data can be used to better adapt CLIP models for zero-shot learning.

*How to generate the data?* Given a pre-trained text-to-image generation model, to synthesize novel samples, the basic (**B**) strategy is to use the label names of the target categories to build the language input and generate a corresponding image. Then, the paired label names and synthesized data can be employed to train the classifier with the feature extractor frozen.

*How to enrich diversity?* Only using the label names as inputs might limit the diversity of synthesized images and cause bottlenecks for validating the effectiveness of synthetic data. Hence, we

| Dataset | Task | CLIP-RN50 | CLIP-RN50+SYN | CLIP-ViT-B/16 | CLIP-ViT-B/16+SYN |
|---|---|---|---|---|---|
| CIFAR-10 | o | 70.31 | 80.06 (+9.75) | 90.80 | 92.37 (+1.57) |
| CIFAR-100 | o | 35.35 | 45.69 (+10.34) | 68.22 | 70.71 (+2.49) |
| Caltech101 | o | 86.09 | 87.74 (+1.65) | 92.98 | 94.16 (+1.18) |
| Caltech256 | o | 73.36 | 75.74 (+2.38) | 80.14 | 81.43 (+1.29) |
| ImageNet | o | 60.33 | 60.78 (+0.45) | 68.75 | 69.16 (+0.41) |
| SUN397 | s | 58.51 | 60.07 (+1.56) | 62.51 | 63.79 (+1.28) |
| Aircraft | f | 17.34 | 21.94 (+4.60) | 24.81 | 30.78 (+5.97) |
| Birdsnap | f | 34.33 | 38.05 (+3.72) | 41.90 | 46.84 (+4.94) |
| Cars | f | 55.63 | 56.93 (+1.30) | 65.23 | 66.86 (+1.63) |
| CUB | f | 46.69 | 56.94 (+10.25) | 55.23 | 63.79 (+8.56) |
| Flower | f | 66.08 | 67.05 (+0.97) | 71.30 | 72.60 (+1.30) |
| Food | f | 80.34 | 80.35 (+0.01) | 88.75 | 88.83 (+0.08) |
| Pets | f | 85.80 | 86.81 (+1.01) | 89.10 | 90.41 (+1.31) |
| DTD | t | 42.23 | 43.19 (+0.96) | 44.39 | 44.92 (+0.53) |
| EuroSAT | si | 37.51 | 55.37 (+17.86) | 47.77 | 59.86 (+12.09) |
| ImageNet-Sketch | r | 33.29 | 36.55 (+3.26) | 46.20 | 48.47 (+2.27) |
| ImageNet-R | r | 56.16 | 59.37 (+3.21) | 74.01 | 76.41 (+2.40) |
| Average | / | 55.13 | 59.47 (+4.31) | 65.42 | 68.32 (+2.90) |

Table 1: **Main Results on Zero-shot Image Recognition.** All results are top-1 accuracy on test set. o: object-level. s: scene-level. f: fine-grained. t: textures. si: satellite images. r: robustness.

leverage an off-the-shelf word-to-sentence T5 model (pre-trained on "Colossal Clean Crawled Corpus" dataset (Raffel et al., 2020) and finetuned on CommonGen dataset (Lin et al., 2019)) to increase the diversity of language prompts and the generated images, namely language enhancement (**LE**), hoping to better unleash the potential of synthesized data. Concretely, we input the label name of each class to the word-to-sentence model which generates diversified sentences containing the class names as language prompts for the text-to-image generation process. For example, if the class label is "airplane", then the enhanced language prompt from the model could be "a white airplane hovering over a beach and a city". The enhanced text descriptions introduce rich context descriptions.

*How to reduce noise and enhance robustness?* It's unavoidable that the synthesized data may contain low-quality samples. This is even more severe in the setting with language enhancement as it may introduce undesired items into language prompts (see Figure A.2 in Appendix for visual examples). Hence, we introduce a CLIP Filter (**CF**) strategy to rule out these samples. Specifically, CLIP zero-shot classification confidence is used to assess the quality of synthesized data, and the low-confidence ones are removed. Besides, as soft-target is more robust than hard-target in countering sample noise, we study whether soft cross-entropy loss (**SCE**, see Sec. C.4 in Appendix) which uses the normalized clip scores as a target could be used to enhance robustness against data noise.

**Experiment Setup**. We select 17 diverse datasets covering object-level (CIFAR-10 and CIFAR-100 ((Krizhevsky et al., 2009), Caltech101 (Fei-Fei et al., 2006), Caltech256 (Griffin et al., 2007), ImageNet (Deng et al., 2009)), scene-level (SUN397 (Xiao et al., 2010)), fine-grained (Aircraft (Maji et al., 2013), Birdsnap (Berg et al., 2014), Cars (Krause et al., 2013), CUB (Wah et al., 2011), Flower (Nilsback & Zisserman, 2008), Food (Bossard et al., 2014), Pets (Parkhi et al., 2012)), textures (DTD (Cimpoi et al., 2014)), satelite images (EuroSAT (Helber et al., 2019)) and robustness (ImageNet-Sketch (Wang et al., 2019), ImageNet-R (Hendrycks et al., 2021)) for zero-shot image classification. For synthetic data amount, we generate 2000 (study of synthetic image number in Appendix Sec. B.3) synthetic images for each class in **B** and **LE**. For **LE**, we generate 200 sentences for each class.

**Main Results:** 1) zero-shot classification results on 17 datasets; 2) study of synthetic data diversity; 3) study of synthetic data reliability; 4) study of model/classifier tuning; 5) study of the behavior of synthetic data for zero-shot classification in the training from scratch settings.

*Synthetic data can significantly improve the performance of zero-shot learning.* Our main studies in zero-shot settings are conducted with CLIP-RN50 (ResNet-50 (He et al., 2016) and CLIP-ViT-B/16 (ViT-B/16 (Dosovitskiy et al., 2020)) as CLIP backbone), and we report results with our best strategy of **LE+CF+SCE**. As shown in Table 1, on 17 diverse downstream zero-shot image classification datasets, we achieve a remarkable average gain of 4.31% for CLIP-RN50 and 2.90% for CLIP-ViT-B/16 in terms of top-1 accuracy. Significantly, on the EuroSAT dataset, we achieve the largest performance boost of 17.86% for CLIP-RN50 in top-1 accuracy. We notice that the performance gain brought by synthetic data varies differently across datasets, which is mainly related to GLIDE's training data distribution. The training data distribution of the text-to-image generation model GLIDE would exhibit bias and produce different domain gaps with different datasets (see Sec. A.2 in Appendix for more analysis).

| Dataset | CLIP | B | | LE | | LE+CF | |
|---|---|---|---|---|---|---|---|
| | | CE | SCE | CE | SCE | CE | SCE |
| CIFAR-10 | 70.31 | 77.39 (+7.08) | 78.23 (+7.92) | 77.20 (+6.89) | 77.55 (+7.24) | 80.01 (+9.70) | **80.06 (+9.75)** |
| CIFAR-100 | 35.35 | 43.99 (+8.64) | 44.25 (+8.90) | 44.08 (+8.73) | 44.91 (+9.56) | 44.55 (+9.20) | **45.69 (+10.34)** |
| EuroSAT | 37.51 | 45.64 (+8.13) | 48.23 (+10.72) | 53.26 (+15.75) | 54.94 (+17.43) | 54.75 (+17.24) | **55.37 (+17.86)** |

Table 2: Ablation study on **Language Enhancement (LE), CLIP-based Filtering (CF), and Soft-target Cross-Entropy (SCE).**

*Language diversity matters.* By introducing more linguistic context into the text input, **LE** helps increase the diversity of synthetic data. As shown in Table 2, **LE** can achieve additional performance gains upon **B** in most cases (0.66↑ on CIFAR-100, 6.71↑ on EuroSAT), which demonstrates the efficacy of **LE** and the importance of synthetic data diversity for zero-shot classification.

*Reliability matters.* While **LE** could help increase the diversity of synthetic data, it also introduces the risks of noisy samples. Observed on CIFAR-10 in Table 2, **LE** sometimes even brings performance drops compared with **B** (0.68% ↓ on CIFAR-10), which may attribute to the noise introduced by enhanced language prompts, *e.g.* the sentence extended from the class name word may contain other class names or confusing objects. Fortunately, with **CF** to filter out unreliable samples, **LE+CF** yields consistent improvement upon **B**. Moreover, **SCE** generally achieves better performance than **CE**, showing its better adaptation to label noise.

*Classifier tuning is enough for CLIP, while tuning with the pre-trained encoder leads to degradation, mainly due to domain gaps.* Here, we investigate if only tuning the final classifier is the optimal solution in our setting with synthetic data. As shown in Table 3, we tune different proportions of the full model parameters on synthetic data for EuroSAT (0.02% corresponds to our default case where only the classifier is tuned), and report the zero-shot performance on the test set of EuroSAT. The best results are obtained by only tuning the classifier, and the performance gradually decreases as we gradually incorporate more parameters in the encoder for optimization, which agrees with the traditional strategy. For understanding why synthetic data may harm pre-trained image encoder, we experiment with real-world data with domain shifts and find they behave similarly to synthetic data (Appendix Sec. B.2), which suggests that domain gap is the main reason for the phenomenon. We argue that synthetic data might have a better chance to overcome domain shifts in comparison with real-world data since we can customize and keep the label space of the synthetic data in line with the down-stream dataset and use strategies during synthesizing to alleviate domain shifts.

| Param Tuned (%) | 0 | 0.02 | 0.04 | 62.50 | 64.06 | 69.53 | 82.81 | 92.19 |
|---|---|---|---|---|---|---|---|---|
| Acc | 37.51 | **55.37** | 55.11 | 55.28 | 54.56 | 54.34 | 53.63 | 52.09 |

Table 3: **Parameters tuned v.s. Accuracy.** Dataset: EuroSAT.

| Real shot | 1 | 16 | 32 | 64 | 80 | 90 | 95 | 100 |
|---|---|---|---|---|---|---|---|---|
| Acc | 2.48 | 10.4 | 14.95 | 21.96 | 24.4 | 25.52 | 27.99 | 29.95 |

Table 4: **Setting when training from scratch.** Dataset: CIFAR-100.

*Synthetic data deliver inferior performance in the training from scratch setting and are much less data-efficient than real data.* To exclude the influence of powerful CLIP initialization in our study of synthetic data, we also conduct a from-scratch setting on the CIFAR-100 dataset, where we optimize a ResNet-50 model from random initialization. Given the label space of the CIFAR-100 dataset, we generate a synthetic dataset of 50k (500 images per class) to train a ResNet-50 model from scratch for image classification. We achieve a performance of 28.74% top-1 accuracy on CIFAR-100 test set, which is much lower than the performance of the pre-trained CLIP model (see Table 1). This might be attributed to the quality and diversity of data. The CLIP model benefits from diverse real-world data. Further, we hope to investigate how many real in-domain training data can match the performance of our 50k synthetic data. As shown in Table 4, training with 95 images per category ($95 \times 100 = 9.5$k) will achieve comparable performance as that of 50k synthetic data. This manifests that synthetic data are not as efficient and effective as real data when solving downstream tasks. It requires around 5 times more data in order to achieve a comparable performance as that of real data. Note that we find further increasing the amount of synthetic data will not deliver further performance gains for the downstream classification task. We expect that further investigations on synthesis quality will bring new opportunities in this area which will be our future work.

**Summary.** Current synthetic data from text-to-image generation models could indeed bring significant performance boosts for a wide range of zero-shot image classification tasks, and is readily applicable with carefully designed strategies such as large-scale pre-trained models. Diversity and

reliability matter for synthetic data when employed for zero-shot tasks. When the model is trained from scratch with synthetic data, synthetic data cannot deliver satisfactory performance and are much less data-efficient and effective for solving the classification task in comparison with real data.

## 3.2 IS SYNTHETIC DATA READY FOR FEW-SHOT IMAGE RECOGNITION?

In this section, we explore the effectiveness of synthetic data for few-shot tasks and how synthetic data impact the performance as more and more shots are included. Also, we design effective strategies to better leverage synthetic data.

**Few-shot Image Recognition.** We adopt the CLIP-based method as the model for few-shot image recognition due to its state-of-the-art performance (Radford et al., 2021). As discussed previously, various prompt learning based methods can be treated as tuning the classifier weights. We thus study how to tune the classifier weights with synthetic data. In an N-way M-shot case, we are given M real images of each test class, where $M \in \{1, 2, 4, 8, 16\}$ in our experiments. With a total of $N \times M$ training samples, we hope to achieve favorable performance on a hold-out test set of the N classes.

**Synthetic Data for Few-shot Image Recognition.** While there have been a few attempts to study how to better adapt CLIP models for few-shot tasks (Zhou et al., 2022b;a; Zhang et al., 2022), they all focus on the model optimization level, and none have explored from the data level. Here, we systematically study whether and how synthetic data can be employed for solving few-shot image recognition tasks.

With the experience from synthetic data for zero-shot tasks, we adopt the best strategy (*i.e.* **LE+CF**) in the zero-shot setting as the basic strategy (**B**). Further, as the few-shot real samples can provide useful information on the data distribution of the classification task, we develop two new strategies leveraging the in-domain few-shot real data for better using synthetic data: 1) Real Filtering (**RF**): given synthetic data of one class $c$, we use the features of few-shot real samples to filter out synthetic images whose features are very close to the features of real samples that belong to other categories different from class $c$; 2) Real guidance (**RG**): we use the few-shot real samples as guidance to generate synthetic images where the few-shot real samples (added noise) replace the random noise at the beginning of the generation to guide the diffusion process (details in Appendix Sec. C.3).

**Experiment Setup**. For datasets, we carefully select 8 image classification datasets from recent works (Zhou et al., 2022b;a; Zhang et al., 2022) that conduct few-shot learning upon CLIP: ImageNet (Deng et al., 2009), Caltech101 (Fei-Fei et al., 2006), Pets (Parkhi et al., 2012), Cars (Krause et al., 2013), Aircraft (Maji et al., 2013), SUN397 (Xiao et al., 2010), DTD (Cimpoi et al., 2014), EuroSAT (Helber et al., 2019). For synthetic image number, we generate 800 (study of synthetic image number in Appendix Sec. B.3) images per class for **RG** method to approximately match the number of images in **B** and **RF**.

**Main Results:** 1) few-shot classification results on 8 datasets; 2) ablation study of training strategy; 3) ablation study of synthetic data generation strategy; 4) ablation study of BN strategy.

*Synthetic data can boost few-shot learning and the positive impact of synthetic data will gradually diminish with the increase of real data shots.* As shown in Figure 1 (results of more datasets are in the Appendix Sec. B.1), with only few-shot real images for training, our implemented **CT w. init** (classifier weights initialized from CLIP text embeddings) performs comparably with the state-of-the-art CLIP tuning methods **Tip Adapter** (Zhang et al., 2022) and **CoOp** (Zhou et al., 2022b). **CT w. Syn** represents our results of applying synthetic data with mix training, real image as guidance, and freezing BN strategies. With the help of generated synthetic data, **CT w. Syn** achieves noticeable performance gains upon **CT w. init**, and achieves a new state-of-the-art few-shot learning performance across different datasets. We argue that for data-scarce few-shot classification, synthetic data could help address the insufficient data problem to boost performance. However, we notice that the boost from synthetic data gradually diminishes as the real shot number increases. We state that the effectiveness of each sample in real data is high since there's no domain gap; in contrast, synthetic data suffer from domain gaps and perform less efficiently. In addition, the positive effects of the few-shot real data may overlap with that of synthetic data. Thus, with the increase of real data, the overlapping becomes serious and the positive impacts of synthetic data are reduced.

*Mix Training fits few-shot learning with synthetic data.* Now that we have two parts of data, *i.e.* few-shot real data and synthetic data, we could either 1) **phase-wise** train on each part of data with two training phases, or 2) adopt **mix training** that simultaneously utilizes two parts of data to update

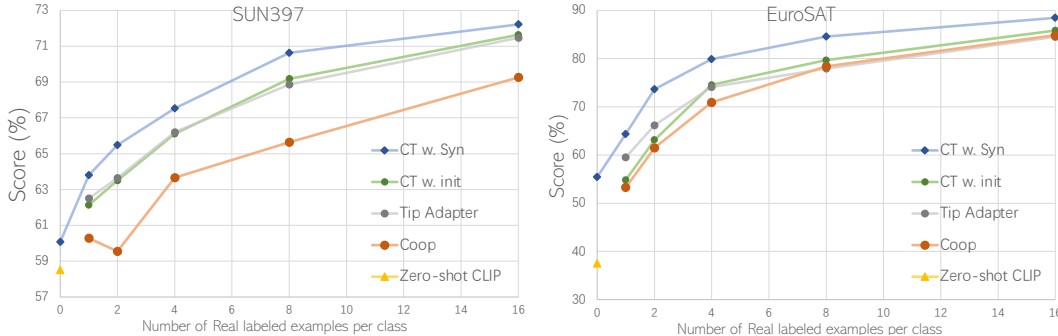

Figure 1: Results for few-shot image recognition. Results on all 8 datasets are provided in Appendix.

the model in each iteration. Details of phase-wise/mix training in Appendix Sec. C.5.2. We provide the results in Table 7: we study on the EuroSAT dataset and use synthetic data generated from the **RG** method; under different shot number settings, mix training performs consistently better than two phase-wise strategies. We suggest that mix training could help learn better classifiers since each part could function as a regularization for the other: synthetic data help alleviate instabilities brought by limited real samples, and real data help address the noise and domain gap of synthetic data.

*Employing real data as guidance can alleviate domain differences and boost performance.* We compare three strategies of synthetic data generation for few-shot tasks. As shown in Table 5, both **RF** and **RG** provide performance gains upon **B** which is the best strategy in the zero-shot setting. This demonstrates the importance of utilizing the domain knowledge from few-shot images for preparing the synthetic data. Further, **RG** significantly outperforms **RF**, yielding the best performance. This shows utilizing real data as guidance of the diffusion process help reduce the domain gap (visual illustrations in the Appendix Sec. B.8).

| B | RF | RG |
|---|---|---|
| 87.1 | 87.33 | 88.47 |

Table 5: Ablation for Basic strategy (**B**), Real Filtering (**RF**), Real Guidance (**RG**) on EuroSAT, 16 shot.

| Train data | Freeze BN? | Test Acc |
|---|---|---|
| Real | | 75.31 |
| Real | ✓ | **85.63** |
| Syn | | 44.73 |
| Syn | ✓ | **55.37** |

Table 6: **Frozen BN works better** for 16-shot settings on EuroSAT.

| M-shot | Phase-wise | | Mix |
|---|---|---|---|
| | syn → real | real → syn | training |
| 1 | 63.01 | 63.32 | **64.36** |
| 2 | 72.24 | 72.85 | **73.62** |
| 4 | 78.88 | 79.21 | **79.88** |
| 8 | 83.64 | 83.99 | **84.57** |
| 16 | 87.10 | 87.44 | **88.47** |

Table 7: **Mix training works better** for few-shot tasks on EuroSAT.

*Frozen BN works better.* Lastly, we investigate batch normalization (BN) strategies for our few-shot settings with synthetic data. As shown in Table 6, for both real and synthetic data, freezing the BN layers yields much better performance. We analyze that for real data, it is hard to get a good estimation of BN statistics when the number of images is limited. As for synthetic data, we attribute this to the statistical difference between different domains. Hence, we freeze BN layers during tuning for few-shot settings.

**Summary.** Synthetic data from text-to-image generation models could readily benefit few-shot learning and achieve a new state-of-the-art few-shot classification performance with strategies we present in this paper. However, the positive impact of synthetic data will diminish as more shots of real data are available which further confirms our previous claim that synthetic data are still not as effective as real data in training classification models.

### 3.3 IS SYNTHETIC DATA READY FOR PRE-TRAINING?

Finally, we study whether synthetic data are effective in large-scale pre-training. We also present effective strategies to better leverage synthetic data for model pre-training.

**Pre-training for Transfer Learning.** Recently, it has become a common practice to first pre-train models on large-scale datasets to obtain a well-trained feature extractor and then fine-tune the models on downstream tasks with labeled data (*i.e.* transfer learning). There have been various successful pre-training methods, including supervised pre-training (Joulin et al., 2016; Li et al., 2017; Mahajan et al., 2018; Sun et al., 2017; Kolesnikov et al., 2020), self-supervised pre-training (Chen et al., 2020a; He et al., 2020; Caron et al., 2020; Grill et al., 2020; Chen & He, 2021; Zbontar et al., 2021; Ye et al., 2019), and semi-supervised pre-training (Xie et al., 2020; Pham et al., 2021).

**Synthetic data for Pre-training.** Since data amount and diversity play important roles in pre-training, we adopt the synthetic data generation strategy **LE** solely to maximize the scale of synthetic pre-training data. We study two settings for generating synthetic data for pre-training: 1) downstream-aware, where we have access to the label space of the downstream task, and thus we generate synthetic data according to the label space of the downstream task; 2) downstream-agnostic, where we have no access to downstream tasks in the pre-training stage, and we turn to a relatively general and diverse label space such as ImageNet-1K. For pre-training methods, we experiment with supervised pre-training and self-supervised pre-training methods.

**Experiment Setup**. We compare synthetic pre-trained models with models of random initialization and models of ImageNet-1K pre-training in terms of their transfer learning abilities. For downstream-aware settings: we conduct supervised pre-training on synthetic data generated according to CIFAR-100 label space and then transfer to CIFAR-100 through finetuning for evaluation.

For downstream-agnostic settings: we perform supervised pre-training and self-supervised pre-training (we adopt Moco v2 (Chen et al., 2020b) framework for its simplicity and reproducibility) on synthetic data generated from ImageNet-1K label space and evaluate the transfer performance by finetuning the pretrained models on a object detection dataset – PASCAL VOC (Everingham et al., 2010). Further, we experiment with ImageNet-2K label space (original ImageNet-1K and another non-overlapping 1K label names randomly selected from ImageNet-21K) to study the factors of data diversity and amount in synthetic pre-training. We use ResNet-50 as the default backbone when not else noted, and also experiment with a ViT-based backbone, *i.e.* DeiT-S (Touvron et al., 2021).

**Results for Downstream-aware settings**. We generate synthetic data of different sizes from CIFAR-100 label space, *i.e.* $1\times$, $2\times$, $3\times$ ImageNet-1K data size, concretely 1.2M, 2.4M, 3.6M. We pre-train the model on the generated synthetic labeled set in a supervised manner, and then perform evaluation after finetuning the model on CIFAR-100. As shown in Table 8, with an equivalent amount of data as that of ImageNet-1K (1.2M), synthetic data for pre-training can largely reduce the gap between training from scratch (78.83%) and ImageNet- pre-trained model (84.50%). Moreover, with $2\times$ and $3\times$ synthetic data, pre-training on synthetic data outperforms ImageNet-1K pre-training with a noticeable margin. In addition, when we initialize the model from ImageNet-1K pre-trained weights and pre-train the model on synthetic data, we obtain extra boosts upon both results.

We conclude that for downstream-aware synthetic pre-training, synthetic data deliver close performance as that of ImageNet-1K pretraining with the same amount of data, synthetic data amount helps improve the results to outperforming ImageNet-1K pre-training, and synthetic pre-training could further benefit from ImageNet-1K pre-training.

**Results for Downstream-agnostic settings**. We first experiment with ImageNet-1K label space with $1\times$ or $2\times$ ImageNet-1K data size, *i.e.* 1.2M/2.4M IN-1K Syn. We perform supervised pre-training and self-supervised pre-training (*i.e.* Moco v2) on the generated synthetic data, and evaluate the pre-training results by transferring to the CIFAR-100 image classification task or the PASCAL VOC detection task. As it is too costly to validate all settings (*e.g.*, it takes more than 1 week to train Moco v2 on 4.0M synthetic data), we select several representative settings of interest to validate the effectiveness of synthetic data without hurting our conclusion.

As shown in Table 10 and 11, with 1.2M IN-1K Syn, both supervised pre-training (79.00%) and self-supervised pre-training (81.55%) could largely approach their IN-1K Real counterparts (super.:81.3%; self-super.:82.44%) and largely outperforms the result without pre-training (66.08%). When increasing the data amount to 2.4M, the transferred results further increase, and the unsupervised pre-training method, *i.e.* Moco v2, performs better in utilizing our synthetic data thanks to its independence of labels, yielding a 82.13% transferred performance which surpasses supervised pre-training on IN-1K Real (81.30%) and is on par with its Moco v2 counterpart at IN-1K Real (82.44%). Next, we expand the label space by adding another 1K categories, producing IN-2K Syn. The enlarged diversity and data amount further bridge the gap between synthetic pre-training results and IN-1K Real pre-training results. Noticeably, the unsupervised pre-trained model Moco v2 (82.29%) largely approaches the IN-1K Real counterpart (82.44%) with negligible performance drop of 0.15%. Furthermore, when initialized from IN-1K Real pre-trained weights, both supervised and self-supervised pre-training improve upon both pure real data and synthetic data for pre-training.

While the above results are all obtained with convolutional-based backbone *i.e.* ResNet50, we further explore with a recent ViT-based backbone *i.e.* DeiT-S (Touvron et al., 2021). Surprisingly,

| Data | pre-trained on IN-1k? | Syn. images amount | | | |
|------|------|------|------|------|------|
| | | 0 | 1.2M | 2.4M | 3.6M |
| (None) | | 78.83 | - | - | - |
| C100 Syn | | - | 83.90 | **85.03** | **85.24** |
| (None) | ✓ | 84.50 | - | - | - |
| C100 Syn | ✓ | - | **84.90** | **85.32** | **85.52** |

Table 8: Results on CIFAR-100 with **downstream-aware supervised pre-training**. C100: CIFAR100.

| Data | pre-trained on IN-1k? | Syn. images amount | | | |
|------|------|------|------|------|------|
| | | 0 | 1.2M | 2.4M | 4.0M |
| (None) | | 69.29 | - | - | - |
| IN-1K Syn | | - | 87.98 | **88.39** | - |
| IN-2K Syn | | - | - | **88.57** | **88.91** |
| (None) | ✓ | - | 88.07 | - | - |

Table 9: Results on CIFAR-100 with **downstream-agnostic supervised pre-training**. Backbone: DeiT-S.

| Data | pre-trained on IN-1k? | Syn. images amount | | | |
|------|------|------|------|------|------|
| | | 0 | 1.2M | 2.4M | 4.0M |
| (None) | | 66.08 | - | - | - |
| IN-1K Syn | | - | 79.00 | 80.00 | - |
| IN-2K Syn | | - | - | 80.54 | 80.72 |
| (None) | ✓ | 81.30 | - | - | - |
| IN-1K Syn | ✓ | - | - | **81.78** | - |
| IN-2K Syn | ✓ | - | - | **81.87** | **81.91** |

Table 10: Results for object detection on PASCAL VOC with **downstream-agnostic supervised pre-training**, all results are reported in $AP_{50}$.

| Data | pre-trained on IN-1k? | Syn. images amount | | | |
|------|------|------|------|------|------|
| | | 0 | 1.2M | 2.4M | 4.0M |
| (None) | | 66.08 | - | - | - |
| IN-1K Syn | | - | 81.55 | 82.13 | - |
| IN-2K Syn | | - | - | 82.22 | 82.29 |
| (None) | ✓ | 82.44 | - | - | - |
| IN-1K Syn | ✓ | - | - | **82.47** | - |

Table 11: Results for object detection on PASCAL VOC with **downstream-agnostic self-supervised pre-training (Moco v2)**, all results are reported in $AP_{50}$.

ViT-based backbone is shown to be more advantageous compared with convolution-based backbone for synthetic pre-training: outperforming ImageNet pre-training results in the downstream-agnostic settings. Equipped with ViT-based backbone, on only 1.2M IN-1K synthetic data, we achieve comparable performance (87.98%) with ImageNet pre-training (88.07%). Further increasing the data amount (88.39%) and label space (88.57%, 88.91%) of pre-training data leads to higher performance than ImageNet pre-training. ViT-based backbones have stronger ability for learning from large-scale data and are more robust (Pinto et al., 2021), and thus could better benefit from synthetic pre-training where data are more noisy and data scale could be easily increased.

**Conclusion**. In terms of transfer abilities, synthetic data from text-to-image generation models show surprisingly promising results for model pre-training, which is comparable to the standard ImageNet pre-training. We conclude our findings as follows:
1. Data amount has positive impacts on synthetic pre-training; performance could be improved by increasing synthetic data size, but would gradually saturate as the amount of data increases.
2. Synthetic data for pre-training is orthogonal to real data for pre-training.
3. For downstream-aware synthetic pre-training, we significantly outperform IN-1K Real (1.2M) pre-training with 2.4M/3.6M synthetic data on CIFAR-100.
4. For downstream-agnostic synthetic pre-training, we achieve comparable results with ImageNet (IN-1k) Real pre-training; self-supervised pre-training performs better than supervised pre-training, and ViT-based backbone performs better than convolutional-based backbone. Besides, increasing the label space size could further improve the performance.

## 4 CONCLUSION

We systematically investigate whether synthetic data from current state-of-the-art text-to-image generation models are readily applicable for image recognition. Our extensive experiments demonstrate that synthetic data are beneficial for classifier learning in zero-shot and few-shot recognition, bringing significant performance boosts and yielding new state-of-the-art performance. Further, current synthetic data show strong potential for model pre-training, even surpassing the standard ImageNet pre-training. We also point out limitations and bottlenecks for applying synthetic data for image recognition, hoping to arouse more future research in this direction.

**Limitations.** In all investigated settings, we observe improved performance as the data amount and diversity (label space) increases. However, due to our limited computational resource, we are not able to further scale up data amount, which may take months to train one model. Besides, we are also not able to investigate larger model sizes and advanced architectures in the current investigation which is also worth exploring in the future. We present more discussions on limitations and future directions in the appendix.

**Acknowledgement.** This work has been supported by Hong Kong Research Grant Council - Early Career Scheme (Grant No. 27209621) and General Research Fund Scheme (Grant no. 17202422). Part of the described research work is conducted in the JC STEM Lab of Robotics for Soft Materials funded by The Hong Kong Jockey Club Charities Trust.

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
