# OpenReview forum: "IS SYNTHETIC DATA FROM GENERATIVE MODELS READY FOR IMAGE RECOGNITION?"
_ICLR.cc/2023/Conference — ICLR 2023 notable top 25%_

### Official Review · Reviewer_NhD1 · 2022-10-22

**Confidence:** 4
**Correctness:** 3
**Technical Novelty And Significance:** 3
**Empirical Novelty And Significance:** 3
**Recommendation:** 6

**Clarity, Quality, Novelty And Reproducibility:**

Overall, the paper reads well, but the quality of the work could be improved as it presents too much detail in a disorganised manner, for example section 3.1 is too long and could benefit from subsections. The experiments are rigorous and the work appears to be reproducible, although the authors are encouraged to provide their working code.

Minor typo:
- The paragraph "This manifests that synthetic data are not as ..." on page 5.




**Strength And Weaknesses:**

Strengths:
- The overall idea aims to investigate the suitability of synthetic datasets for performing image recognition tasks, which can be valuable in a data-scarce environment
- The language enhancement method for expanding words into sentences is a useful method for generating diverse sentences, which can increase the diversity of synthetic datasets generated
- The real guidance method appears to be novel in helping the model bridge the domain gap between synthetic and real-world datasets, improving the performance of the model
- The paper shows a significant improvement over the model CLIP and demonstrates that synthetic datasets can indeed increase the performance of the model

Weekness:
- The paper mentions some previous work that investigated similar questions, albeit on a smaller scale. A comparison of their results with these methods could further strengthen their work
- The paper uses 17 datasets to evaluate their method and compares it to the CLIP model. Admittedly, 17 datasets is quite impressive in itself, but the CLIP paper evaluated 30 datasets. Can the authors explain why the remaining 13 datasets were excluded in their analysis?
- In the paper, 2000 samples per class were synthesised to fine-tune the CLIP model and an average gain of 4.31 was obtained over the original model. Were the 2000 samples chosen arbitrarily, or was the model fine-tuned using different synthetic samples and 2000 determined to be the magic number
- On page 3 "...Here, we adopt a simple tuning method, Classifier Tuning (CT) (Wortsman et al., 2022), which directly tunes a classifier attached to CLIP image encoder and initializes the classifier weights with pre-trained text embedding." The description of how the classifier tuning (CT) method works does not match my understanding of the paper (Wortsman et al. 2022), which describes the WiSE- FT method as a weighted linear interpolation between the original zero-shot and fine-tuned models. Perhaps the authors could elaborate on how CT uses the pre-trained text embeddings to initialise the classifier weights, or refer me to the relevant paragraph in the Wortsman et al. 2022 paper that does so.
- On page 6, It is not clear to me how phase-wise training works, nor was the method of mix training well explained, making it difficult to understand the performance gain from the procedure shown later in Table 7. It would be nice if the authors would refer to the relevant literature where these methods are presented or explain in more detail how they work in the appendix

**Summary Of The Paper:**

The paper investigated the use of synthetic data generated from state-of-the-art text-to-image generation model (GLIDE) to improve image recognition. This was approached from two angles: Fine-tuning the state-of-the-art model for zero-shot learning (CLIP) on synthetic datasets, considering both zero-shot and few-shot settings, and then comparing the performance of pre-trained weights from synthetic datasets with real data to investigate the suitability of synthetic datasets for transfer learning. In addition, several strategies were introduced to improve the diversity and quality of the generated synthetic datasets.

**Summary Of The Review:**

The results of this work are insightful and contribute to a better understanding of the capabilities of synthetic datasets. They also shed light on the challenges associated with sample diversity and quality. Depending on the task, the synthesised data from the text-image generation model could simply be full of noise, making this method unlikely to be very useful in practise.

---

> ### Author Response · Authors · 2022-11-18
> **Author Feedback for Reviewer NhD1: Part 2**
>
> **Q4: Please elaborate more on Classifier Tuning.**
>
> A4: **Clarification of Classifier Tuning (CT).**
> We added the clarification in our updated paper. For Classifier Tuning (CT), we refer to the standard fine-tuning in paper [1] that only tunes a classifier while fixing the image encoder (refer to Sec. 2 subsections “Zero-shot models and CLIP” and the “ fine-tuning only a linear classifier” version in Sec. 3 “Step 1” in [1] ), but not the weight-space ensembling method. Concretely, for a given zero-shot image classification task, we input the label names with prompts into CLIP’s text encoder to obtain the text features, and use the text features as the initialization weights of the classifier appended to the image encoder for the zero-shot task. For all variants of CT mentioned in our paper, we use the text features as initialization. CT w. init and CT w. syn are namings in the few-shot settings to distinguish between tuning only with real data and tuning with both real and synthetic data.
>
> [1] Mitchell Wortsman et al. Robust fine-tuning of zero-shot models. In Proceedings of the IEEE/CVF Conference on Computer Vision and Pattern Recognition, 2022.
>
> **Q5: Please provide details for phase-wise training and mixed training.**
>
> A5: We **added the implementation details** of phase-wise training and mix training in the appendix. For phase-wise training, we utilize synthetic data and real data in two different phases, and the order of using synthetic data and real data also yields two variants. For syn->real, we first tune the classifier on synthetic data for 30 epochs, and then tune it on real data for 30 epochs; and change the order for real->syn. For mix training, in each training iteration, we get a batch of real data input into the model and obtain the loss value of real part data, and also get a batch of synthetic data input into the model and get a loss value of synthetic part data, and then add two loss values as the final loss to do backpropagation.
>
> **Q6: Depending on the task, the synthesized data from the text-image generation model could simply be full of noise, making this method unlikely to be very useful in practice.**
>
> A6: We provide synthesized images for different datasets (i.e., CIFAR10, Caltech101, Cars, ImageNet-Sketch, DTD) **in Appendix Figure A.5**. All images are randomly chosen rather than human-picked. Each row consists of images of the same class. We observe that for most datasets, synthesized images from the GLIDE model are of high quality, but there also exist cases in that many unsatisfactory examples are generated, such as the DTD datasets.
>
> This is a limitation of the current text-image generation model that it may produce images of low quality for certain tasks, which may be due to the domain gap between the training data of the generation model and the task. However, with the study of future text-image generation models, the quality of synthesized images is potentially growing higher constantly. Besides, the relatively lower-quality images could be used for pre-training tasks for better improving the performance of different downstream tasks.

---

> > ### Comment · Reviewer_NhD1 · 2022-11-24
> > **Response**
> >
> > I thank the authors for their response and for revising their manuscript.

---

> > > ### Author Response · Authors · 2022-11-25
> > > **Author Feedback**
> > >
> > > Dear Reviewer NhD1,
> > >
> > > Thanks for your response!
> > > Is there anything more you would like to discuss or do you still have some confusion or concerns?
> > >
> > > Best regards, Paper 220 Authors

---

> > > > ### Comment · Reviewer_NhD1 · 2022-11-29
> > > > **Response**
> > > >
> > > > In general, this is a well-written paper that has been further improved by revision, and I commend the authors' efforts. However, I would encourage the authors to consider restructuring their results section, as I believe the current presentation hides many useful insights between texts. I found that some findings would have benefited from more discussion. For example, at the end of Section 3.2, the authors conclude that a model trained from scratch with synthetic data cannot perform satisfactorily, but why is this? If this is a problem with the diversity of images, are there some images from the real data that carry more signals than others, such that their absence in the synthetic training process led to the poor performance? I realise that these are difficult questions, but they are important nonetheless so that we can develop a better understanding of the use of synthetic datasets for downstream tasks and their limitations. I raised my score to 6 because I find the work insightful and some of the techniques used to improve zero- and few-shot learning experiments impressive -- and I think this work is beneficial to the scientific community's understanding of the utility of synthetic datasets for downstream tasks. I now tend towards accept and I thank the authors for their response.

---

> > > > > ### Author Response · Authors · 2022-11-29
> > > > > **Author Feedback**
> > > > >
> > > > > Dear Reviewer NhD1,
> > > > >
> > > > > Thanks for your response and suggestions! We will try our best to restructure our paper, and address these questions in our next version. Also, for the raised questions that may still be difficult to answer for now, we hope we could arouse more research interest in using synthetic data and more researchers could join to explore ideas in this direction.
> > > > >
> > > > > Best regards, Paper 220 Authors

---

> ### Author Response · Authors · 2022-11-18
> **Author Feedback for Reviewer NhD1: Part 1**
>
> Dear Reviewer NhD1,
>
> Thank you for appreciating our approach. We will address your comments below.
>
> **Q1: Why not compare with previous works that investigated similar questions on a smaller scale?**
>
> A1: **Comparing the text-to-image generation model with the previous GAN-based generation model is not the focus of our paper.**
> Previous works explore upon GAN-based model, and the images generated are limited to the label space of the training data, e.g., GAN trained on ImageNet data could only generate data of ImageNet label space. In contrast, **we are the first that aim to** examine whether the newly arisen text-to-image generation models are ready for image recognition tasks, and we could generate synthetic images of arbitrary label space. Note that we do not aim to compare the text-to-image generation model with the previous GAN-based generation model, hence we do not include them for comparison.
>
> **Q2: Why not use all 27 datasets in the CLIP paper?**
>
> A2: **For datasets in zero-shot settings**, we follow previous works [1, 2] that used 11 datasets, and we exclude UCF101 since GLIDE excludes generating ‘person’ related content for privacy issues. Besides, we add another 7 popular datasets for more comprehensive evaluation. We do not conduct on all CLIP’s 27 datasets since our computing resources are limited, and we believe our 17 datasets are already enough to study the effectiveness of synthetic data for zero-shot settings.
>
> [1] Kaiyang Zhou et al. Learning to prompt for vision-language models. International Journal of Computer Vision, 2022.
> [2] Renrui Zhang et al. Tip-adapter: Training-free adaption of clip for few-shot classification. arXiv preprint arXiv:2207.09519, 2022.
>
> **Q3: Why is the number of synthetic images 2000?**
>
> A3: **Added study of synthetic image number in zero/few-shot settings.**
> Here, we provide the study for the number of synthetic images in the zero-shot and few-shot settings.
> Firstly, for zero-shot settings, we experiment with 1000/2000/4000 images per class on CIFAR-10. As shown **in Table A.2 in our Appendix**, we found 2000 to be a sufficient number while further increasing the number to 4000 only provides limited gains.
> Second, for few-shot settings, we study the settings of Eurosat dataset with 16 shot real images, and vary the number of synthesized images for each class from 400~1600. As shown **in Table A.3 in our Appendix**, we found 800 synthetic images for each class to be a good choice between performance and cost.
>
> For both settings, increasing the amount of training data further beyond a certain amount will not bring significant performance gains. The reasons can be attributed to the diversity and quality of data. We found that with the increase in the training data amount, the diversity of the data might not be scaled in a similar manner. Many redundant and similar samples will appear with the increase in data amount. Effective approaches to increase data diversity and quality will help further improve model performance.
> For the few-shot setting, the performance reaches a high value after using a small amount of synthetic data (400-shot). This is because the existence of real data provides strong guidance for training the classifier. And, the positive impacts of synthetic data are reduced where a small amount of synthetic data are sufficient to learn a good classifier.

---

### Official Review · Reviewer_WkSM · 2022-10-23

**Confidence:** 4
**Correctness:** 4
**Technical Novelty And Significance:** 2
**Empirical Novelty And Significance:** 2
**Recommendation:** 6

**Clarity, Quality, Novelty And Reproducibility:**

The paper writing is very clear. The idea is interesting but not a breakthrough. The experiment results are through, and the results are somehow expected. However, it is hard to extensively evaluation on all recent foundational models.

**Details Of Ethics Concerns:**

Training with synthetic data may introduce bias issues.

**Strength And Weaknesses:**

+ show state-of-the-art results on zero-shot and few-shot recognition on a range of datasets
+ the first systematic study on using generative synthetic data for classification
- foundational models and approaches are limited to GLIDE and CLIP
- improvement over CLIP baseline not very large
- time complexity is high

**Summary Of The Paper:**

This paper investigates using synthetic data from generated with GLIDE model for image recognition based on CLIP embedding in zero-shot and few-shot settings. It is shown that synthetic data are beneficial for classifier learning. The synthetic data also show great potential for model pre-training.

**Summary Of The Review:**

This paper is interesting but not providing deep insights. More follow up work may be needed.

---

> ### Author Response · Authors · 2022-11-18
> **Author Feedback for Reviewer WkSM**
>
> Dear Reviewer WkSM,
>
> Thank you for appreciating our approach. We will address your comments below.
>
>
> **Q1: foundational models and approaches are limited to GLIDE and CLIP.**
>
> A1: **Emphasis on our aim and added experiments of ViT synthetic pre-training.**
> We aim to examine whether the current state-of-the-art text-to-image models could benefit image recognition instead of studying foundation models. We emphasize that our work could be widely followed for different zero/few-shot methods and pre-training methods. Note that we use CLIP for zero/few-shot settings since it is the state-of-the-art method for zero-shot and few-shot image recognition, and our synthesized images targeted for a certain zero/few-shot task could surely be utilized by other zero/few-shot methods. In our pre-training settings, we attempted supervised pre-training and self-supervised pre-training with ResNet-50, and we argue that different pre-training methods and backbones could be used for the synthetic pre-training setting.
>
> Here, we add experiments of ViT-based backbone: we supervised pre-train DeiT-S on different sets of synthetic data from random initialization and the transfer results on CIFAR100 are shown **in Table 9 in our paper**. Surprisingly, **ViT-based backbone is shown to be more advantageous compared with the convolution-based backbone for synthetic pre-training**: outperforming ImageNet pre-training results in the downstream-agnostic settings. Equipped with the ViT-based backbone, on only 1.2M IN-1K synthetic data, we achieve comparable performance  (87.98%) with ImageNet pre-training (88.07%). Further increasing the data amount (88.39%) and label space (88.57%, 88.91%) of pre-training data leads to higher performance than ImageNet pre-training.
>
> **Q2: the improvement over CLIP baseline not very large.**
>
> A2: **Analysis of improvement over CLIP baseline.**
> For zero-shot tasks, we show synthetic data could bring a large boost as much as 17.86% on Eurosat, and averagely 4.31% on 17 diverse datasets with CLIP-RN50. However, we indeed found some datasets where we only achieve marginal gains, which we attribute to GLIDE’s training data distribution. The training data distribution of the text-to-image generation model GLIDE would exhibit bias and produce different domain gaps with different datasets. Detailed explanations are provided **in Sec. A.2 in Appendix**. For few-shot tasks, when real data are introduced, the boost from synthetic data gradually decreases which we acknowledge as a limitation in our paper. We attribute this to the inefficiency of synthetic images due to domain gaps compared with in-domain real images. Besides, the positive effects of the few-shot real data may overlap with that of synthetic data.
>
> **Q3: time complexity is high.**
>
> A3: 1. Compared with the cost of building up a new real-world dataset, **our cost of generating synthetic data from generative models is largely reduced**.
>
> 2.(1) **For zero-shot and few-shot tasks**, we only employ a median size synthetic dataset, tune a light-weight classifier, and use a short 30 epoch schedule, the cost additionally added by learning from synthetic data is actually quite low, e.g., **several (<10) minutes** on just **one single GPU**.
>
> 2.(2) **For pre-training**, we admittedly need more amount of synthetic data to achieve comparable performance with real data, which is a limitation of current synthetic data.
>
> 3.**For inference**, the time cost remains the same as the synthetic data is only applied during training.
>
> **Q4: The idea is interesting but not a breakthrough, not providing deep insights.  The results are somehow expected.**
>
> A4: **We argue that we provide several inspiring and unexpected findings:**
>
> **1. For zero-shot settings**, we show the efficacy of using language enhancement for improving diversity and using CLIP confidence selection for improving reliability.
>
> **2. For few-shot settings**, we propose to use real few-shot images as guidance in text-to-image generation for reducing domain gap and use mix training when synthetic data and real data are both at hand.
>
> **3. For pre-training settings**, we demonstrate that synthetic pre-training could produce comparable or even better performance than the standard ImageNet pre-training; and synthetic data for pre-training favors self-supervised pre-training compared with supervised pre-training, ViT-based backbone compared with convolution-base backbone.
>
> (More analysis can be referred to Sec. 3 in our paper.)

---

> ### Author Response · Authors · 2022-12-07
> **Author Feedback**
>
> Dear Reviewer WkSM,
>
> Thank you for your time and efforts in evaluating our paper. Is there anything more you would like to discuss or do you still have some confusion or concerns?
>
> Best regards, Paper 220 Authors

---

### Official Review · Reviewer_xQXo · 2022-10-25

**Confidence:** 3
**Correctness:** 3
**Technical Novelty And Significance:** 2
**Empirical Novelty And Significance:** 3
**Recommendation:** 6

**Clarity, Quality, Novelty And Reproducibility:**

The paper is clear, has good quality and the experiments seems to be reproducible.

**Strength And Weaknesses:**

The problem is interesting, and could be potentially useful in real world applications.

The paper conduct different experiments to investigate how do the synthetic data from generative models contribute to the image recognition problem under different settings, including zero-shot, few-shot, pre-training then transfer learning settings.

Experimental results are clear and seems to be reasonable.

Although the paper is an experiment-based paper which aims to reveal some interesting applications, deeper investigation are expected rather than only conducting experiments. Specifically, what's the reason behind the performance differences (different levels of improvements across datasets are observed, e.g. accuracy is only improved by 0.01% on Food, while the improvement on EuroSAT is 17.86%). If we find a reasonable or intuitive explanation, can we use it to further improve the results by controlling the generation or augmentation process.

It would be nice if the author could provide some examples of synthetic and ground-truth data. Since the experiments are conducted on many different datasets, intuitively, the more similar synthetic data to ground-truth data, the better performance mode can get, but is it
 true? Some examples along with Table 1 and 2 may help us to better understand why synthetic data contributes to image recognition.

Although zero-shot, few-shot problems are important, how about fully-supervised setting? Will the synthetic data benefits the model when lots of ground-truth data are provided or it will be harmful to the model?

**Summary Of The Paper:**

This paper investigates an interesting problem, whether text-to-image generation model can help image recognition through synthetic data.

**Summary Of The Review:**

The paper investigates how will text-to-image generation model helps image recognition. Because of the clarity and meaningful experiments, I lean towards acceptance, but I do hope to see more results or discussions from the authors.

---

> ### Author Response · Authors · 2022-11-18
> **Author Feedback for Reviewer xQXo**
>
> Dear Reviewer xQXo,
>
> Thank you for appreciating our approach. We will address your comments below.
>
> **Q1: what's the reason behind the performance differences in the zero-shot settings? If we find, how to improve the results?**
>
> A1: **Performance differences are mainly related to GLIDE’s training data distribution. More efforts in tuning the image generator could be beneficial.**
> The training data distribution of the text-to-image generation model GLIDE would exhibit bias and produce different domain gaps with different datasets. While we do not have direct access to GLIDE’s training data, we may use synthesized images from GLIDE to explore. However, how to measure data and its relation to the performance for different tasks in a quantitative manner is still an open problem to our best knowledge as different tasks may have different levels of difficulties and data will impact the performance in different manners. An empirical thought is related to the synthesized image quality (e.g., synthesized images for DTD dataset are relatively noisy as shown **in Figure A.5 in Appendix**, where we only achieve 0.96% gains), domain gaps (synthesized images are usually object-centric and exhibit large domain difference with scene-level dataset SUN397, where only 1.56% boost achieved), and task difficulties (e.g., ImageNet is of high difficulty where only 0.45% gains are obtained).
>
> For further improving the results, we suggest that it would be interesting to investigate how to generate in-domain images by tuning the image generator together with the lateral model for downstream tasks. This could potentially better reduce the domain gap to the downstream task.
>
> **Q2: Please provide examples of synthetic and ground-truth data. What’s the relation between domain gap/data similarity and performance gains?**
>
> A2: **Reduced domain gaps could help boost performance from both qualitative and quantitative results.**
> For understanding the domain gaps (i.e., the similarity between synthetic data and ground-truth/real data),
>
> **1) qualitatively**, we provide visual examples in our Appendix (**see Figure A.3**). On EuroSAT dataset, we show examples of synthesized images of the “highway or road” class by different methods (B, LE, LE+CF). We could see that LE could help increase the diversity but may introduce noisy samples, but LE+CF could further select images with higher class fidelity and yield synthetic data with reduced domain gaps.
>
> **2) quantitatively**, we could see from the FID **in Table A.5 in our Appendix**, which measures the distance between ground-truth data and different sets of synthetic data.
>
> **3) consequently**, Combining visual examples and FID scores with the results in **Table 2 in our paper**, we conclude that a reduced domain gap between synthetic data and ground-truth data could be beneficial for improving performance.
>
> **Q3: How would synthetic data help with a fully-supervised setting? Will it be beneficial or harmful?**
>
> A3: **Explanations about fully-supervised settings.**
> We don't study the fully supervised setting because we obtain little performance improvement when the real shot exceeds 8~16 in our few-shot settings. As the effects of real data will overlap with synthetic data, we expect there will be very limited gains in the fully-supervised setting. This is a limitation of synthetic data. However, we believe with the quality and diversity of synthesized data increase, the positive impacts of synthetic data will be amplified even in this challenging scenario.
>
> Moreover, for fully-supervised tasks, we turn to a different angle: the pre-training and transfer setting, where we generate massive synthetic data for pre-training and then transfer it to the fully-supervised task.

---

> > ### Comment · Reviewer_xQXo · 2022-11-26
> > **Thank the authors for the response**
> >
> > I have read the response to my questions and other reviewers' questions.
> > I do not have further questions.
> > I would like to keep the score, lean towards acceptance.

---

### Official Review · Reviewer_z3p6 · 2022-10-26

**Confidence:** 4
**Correctness:** 3
**Technical Novelty And Significance:** 2
**Empirical Novelty And Significance:** 3
**Recommendation:** 8

**Clarity, Quality, Novelty And Reproducibility:**

Please clarify that the zero-shot classification performance of CLIP-Res50 in the paper is slightly lower than that of the original CLIP paper (~5% in CIFAR-10, CIFAR-100).
In the ablation studies on zero-shot (B, LE, CF) and few-shot (B, RG, RF) classification, qualitative results with FID and classification accuracy with a pre-trained classifier can help to check the diversity and class fidelity.


**Strength And Weaknesses:**

Strength -

The paper is overall well-written and easy to follow.
The extensive experiments using the synthetic dataset support the main claim.
They proposed a novel approach that utilizes text-to-image pre-trained models for a classification dataset generation that does not use any image and label for zero-shot and few-shot classification.
They showed that tuning the classifier on the pre-trained CLIP model with a synthetic dataset increases zero-shot classification performance.
Supervised and self-supervised learning methods are used in pre-training for transfer learning and boosted the classification CIFAR-100 and object detection on PASCAL VOC 2012.


Weaknesses -

They showed the enhanced results only when tuning on the synthetic dataset. However, in some cases, the synthetic dataset could harm the pre-trained image encoder (e.g., Scratch training on CIFAR-100 with a synthetic dataset and Table 3). The analysis of failed experiments when mixing the real and the synthetic datasets is also an essential part of this systematic paper and can help future research.

While tuning CLIP with synthetic dataset does not bring performance degradation, the original CLIP is one of the most important parts in zero-shot and few-shot classification. However, Classifier Tuning (CT), and CT w. init are not explained well in the paper. Please explain them clearly in the paper.

A main concern of the paper is that the authors conducted experiments mainly with CLIP-Res50. To be more specific, there are several variants of CLIP for zero-shot classification, such as CLIP-ViT-S, which shows lower FLOPS and high performance.
Furthermore, several variants of CLIP show better performance than the results of this paper, which requires additional training with the generated dataset (e.g., CLIP-ViT-B/16, the base model of Vision Transformer has 91.6, 68.7, and 54.1 zero-shot top-1 accuracy on CIFAR-10, CIFAR-100, and EuroSAT, respectively). Therefore, the paper needs to show the scalability to justify that the generated dataset can improve zero-shot classification regardless of the model’s capacity and achieve state-of-the-art zero-shot classification.

In the pre-training for the transfer learning section, changing the number of synthetic images helps to understand how the synthetic dataset improves the performance. In this respect, further experiments about the number of synthetic images in zero-shot and few-shot classification are necessary to support that the synthetic dataset can help the classification and reveal the limitation of the synthetic dataset. However, the current manuscript does not include experiments for zero-shot and few-shot classification


**Summary Of The Paper:**

This work synthesized classification datasets motivated by high-quality images generated by the pre-trained text-to-image model (GLIDE).
They used the dataset on three tasks: 1) zero-shot classification, 2) few-shot classification, and 3) pre-training for transfer learning.
They tuned the classifier of pre-trained CLIP leveraging the synthesized image-label dataset and improved zero-shot and few-shot classification tasks.
They proposed several simple techniques to build the synthesized dataset with appropriate ablation studies for improving diversity and class fidelity.
Pre-training for transfer learning experiments train models with ImageNet class-based synthesized dataset in a supervised and self-supervised manner.
The pre-training performance nearly reached the ImageNet1K pre-trained model and showed slightly better performance using the ImageNet pre-trained model as a starting point.


**Summary Of The Review:**

They fully leverage the open-source pre-trained text-to-image model to generate the synthetic classification dataset.
Extensive empirical studies support the claim that synthetic datasets can help zero-shot, few-shot classification, and pre-training for transfer learning.
However, the authors did not elaborate sufficiently on the cases when the synthetic images harm tuning the classifier. Specifically, the effect of the proportion of synthesized images are not explained in the paper, which could be the limitation of using the synthetic dataset.
Furthermore, since the authors mainly conducted experiments with CLIP-Res50, it is highly recommended for the authors to include experiments with variants of CLIPs to demonstrate the scalability of the proposed approach.

---

> ### Author Response · Authors · 2022-11-18
> **Author Feedback for Reviewer z3p6: Part 3**
>
> **Q7: Please clarify the zero-shot classification performance differences.**
>
> A7: For all results, we use the officially released CLIP model from {https://github.com/openai/CLIP}. For our original paper, we conduct experiments using simple prompts “a photo of a [CLASS]” or “a photo of a [CLASS], a type of [dataset type]” (for fine-grained tasks) to better evaluate the performance gains from the data itself by excluding the influence of tuned prompts. However, in the original CLIP paper, they use elaborately designed prompt ensembles for each dataset. This is the main reason that our baseline is lower than that of the original paper.
>
> We also provide the results of using CLIP paper’s prompts for 13 datasets (only these 13 datasets out of 17 have reported results from CLIP’s paper) **in Table A.4 in our appendix**. After using the same prompts as the CLIP's paper, the averaged performance on 13 datasets increased from 56.14% to 56.33%, closer to 57.03% of the CLIP's reported results. There are still slight performance differences after using the same prompts, which we suspect to be a small reproduction problem of CLIP since we could match the reported zero-shot results from CoOp [1] and Tip-adapter [2].
>
> We observe that on some datasets we achieve higher results than CLIP reported results (i.e.., CIFAR100, ImageNet, SUN397, Birdsnap, Flower, Pets) and on others, we achieve lower results (i.e., CIFAR10, Caltech101, Aircraft, Cars, Food, DTD, EuroSAT). The averaged zero-shot performance is similar (56.33% v.s. 57.03%) and the performance boost from synthetic data of two different types of prompt is also similar (averaged performance boost on 13 datasets: 3.85% v.s. 4.17%). We argue the slight differences in performance do not affect the investigation of synthetic data.
>
> [1] Kaiyang Zhou et al. Learning to prompt for vision-language models. International Journal of Computer Vision, 2022.
> [2] Renrui Zhang et al. Tip-adapter: Training-free adaption of clip for few-shot classification. arXiv preprint arXiv:2207.09519, 2022.
>
> **Q8: Please provide FID and classification accuracy with a pre-trained classifier.**
>
> A8: Frechet Inception Distance (FID) is a metric that calculates the distance between feature vectors from real and generated images, and lower scores have been shown to correlate well with higher-quality images. Here, we provide the FID scores and classification accuracy with a pre-trained CLIP ViT-B/16 model for measuring the diversity and quality of synthesized images. We study with the Eurosat dataset. For the FID score, we do not have access to GLIDE training data, and thus we calculate FID between the downstream task ground-truth data and synthetic data generated by different strategies. As shown **in Table A.5 in our Appendix**, LE could largely reduce FID and provide large diversity while hurting class fidelity. After combing with CF, FID is further reduced and we also achieve a higher class fidelity. For few-shot settings, RG could largely reduce FID and yield the best class fidelity.

---

> > ### Comment · Reviewer_z3p6 · 2022-11-26
> > **Response**
> >
> > I sincerely appreciate the authors' responses and the tremendous experiments with analysis.
> > Most of my concerns have been addressed.
> > Specifically, the added results of the CLIP-ViT-B/16 backbone for zero-shot settings are impressive.
> > Now I am convined of the scalability of the proposed method.
> > The Classifier Tuning method has been clarified.
> > Therefore I would like to raise my score to '8: Accept'.

---

> ### Author Response · Authors · 2022-11-18
> **Author Feedback for Reviewer z3p6: Part 2**
>
> **Q4: Please clarify Classifier Tuning (CT).**
>
> A4: **Clarification of Classifier Tuning (CT).**
> We added the clarification in our updated paper. For Classifier Tuning (CT), we refer to the standard fine-tuning in paper [1] that only tunes a classifier while fixing the image encoder (refer to Sec. 2 subsection “Zero-shot models and CLIP” and the “ fine-tuning only a linear classifier” version in Sec. 3 “Step 1” in [1] ), but not the weight-space ensembling method. Concretely, for a given zero-shot image classification task, we input the label names with prompts into CLIP’s text encoder to obtain the text features, and use the text features as the initialization weights of the classifier appended to the image encoder for the zero-shot task. For all variants of CT mentioned in our paper, we use the text features as initialization. CT w. init and CT w. syn are namings in the few-shot settings to distinguish between tuning only with real data and tuning with both real and synthetic data.
>
> [1] Mitchell Wortsman et al. Robust fine-tuning of zero-shot models. In Proceedings of the IEEE/CVF Conference on Computer Vision and Pattern Recognition, 2022.
>
> **Q5: Please add CLIP-VIT-B/16 results in zero-shot settings to show the scalability.**
>
> A5: **Notable performance gains are observed in added experiments for CLIP-VIT-B/16.**
> We add experiments of CLIP VIT-B/16 in the zero-shot settings to show the scalability. We use simple prompts (“a photo of a [CLASS]” or “a photo of a [CLASS], a type of [dataset type]” for fine-grained tasks) to better evaluate the performance gains from the data itself by excluding the influence of tuned prompt methods. As shown **in Table 1 in our updated paper**, on all 17 datasets, we observe noticeable gains across different datasets, with an average boost of 2.9. The boost is relatively smaller than CLIP R50, which is expected as boosting performance upon a stronger model is naturally harder.
>
> **Q6: Lack of study of synthetic image number in zero/few-shot setting.**
>
> A6: **Added study of synthetic image number in zero/few-shot settings.**
> Here, we provide the study for the number of synthetic images in the zero-shot and few-shot settings.
> Firstly, for zero-shot settings, we experiment with 1000/2000/4000 images per class on CIFAR-10. As shown **in Table A.2 in our Appendix**, we found 2000 to be a sufficient number while further increasing the number to 4000 only provides limited gains.
> Second, for few-shot settings, we study the settings of the Eurosat dataset with 16 shot real images, and vary the number of synthesized images for each class from 400~1600. As shown **in Table A.3 in our Appendix**, we found 800 synthetic images for each class to be a good choice between performance and cost.
>
> For both settings,  increasing the amount of training data further beyond a certain amount will not bring significant performance gains. The reasons can be attributed to the diversity and quality of data. We found that with the increase in the training data amount, the diversity of the data might not be scaled in a similar manner. Many redundant and similar samples will appear with the increase in data amount. Effective approaches to increase data diversity and quality will help further improve model performance.
> For the few-shot setting, the performance reaches a high value after using a small amount of synthetic data (400-shot). This is because the existence of real data provides strong guidance for training the classifier. And, the positive impacts of synthetic data are reduced where a small amount of synthetic data are sufficient to learn a good classifier.

---

> ### Author Response · Authors · 2022-11-18
> **Author Feedback for Reviewer z3p6: Part 1**
>
> Dear Reviewer z3p6,
>
> Thank you for appreciating our approach. We will address your comments below.
>
> **Q1: Why synthetic dataset may harm the pre-trained image encoder?**
>
> A1: **Domain gaps are the main causes for harming the pre-trained image encoder when it is finetuned for a specific downstream task.**
> We hypothesize that the slight degradation of model performance (Table 3) when tuning the pre-trained image encoder is caused by the domain gaps between synthetic data and real data.  Note that the GLIDE model is trained on images with distribution different from the downstream datasets.  Besides, downstream classification tasks are more prone to being influenced by domain gaps in comparison with pre-training as they are directly used for evaluation.
>
> To gain more insights into this aspect, we experiment with real-world data with domain shifts and find they behave similarly to synthetic data.
> Specifically, we conduct experiments on the pre-trained CLIP model and study two scenarios (Classifier tuning only, and end-to-end finetune) for the downstream Image-Net classification task. These experiments on conducted on three data sources:   1) In-domain real data: ImageNet data is used for training; 2) real-world data with domain shifts: ImageNet-Sketch data which has the same label space with ImageNet but exhibits a domain shift towards sketch is used for training; and 3) Our synthetic data: synthetic data from GLIDE are used for training.  **The results are shown in Table A.1 in our Appendix**. It can be clearly observed that real-world data with domain shifts behave similarly to synthetic data (i.e. end-to-end tuning causes slight harm to the pre-trained encoder.) while the performance of in-domain data will benefit from end-to-end tuning. This suggests that the domain gap is the main reason for harming the pre-trained image encoder.
>
> Besides, **Synthetic data might have a better chance to overcome domain shifts in comparison with real-world data** since we can customize and keep the label space of the synthetic data in line with the downstream dataset. First, synthetic data can be made to tailor to a specific label space that the downstream task requires and reduce category shifts, which yet might be very challenging for real-world data. Second, a small amount of real data can be leveraged to guide the data synthesis process to further alleviate domain shifts (namely Real Guidance in our few-shot settings), which is also worth future in-depth explorations.
>
> **Q2: Please provide analysis for failed experiments when mixing the real and the synthetic datasets.**
>
> A2: **The diminished positive impacts of synthetic data when mixed with real data.** We observe that the positive impacts of synthetic data diminished as the increase of real data in the few-shot experiments. As real data do not suffer from any domain gaps in solving the downstream classification task, the effectiveness and impact of each sample to obtain a good classifier are high. In contrast, synthetic data naturally suffers from domain gaps. Thus the importance of each sample is not as high as real samples. In addition, the positive effects of the few-shot real data may overlap with that of synthetic data. With the increase of real data, the overlapping becomes serious and the positive impacts of synthetic data are reduced.
>
> **Q3: “The original CLIP is one of the most important parts in zero-shot and few-shot classification”.**
>
> A3: **More analysis on inferior performance in the training-from-scratch setting. In comparison with the CLIP pre-trained model, the major performance degradation might be attributed to the quality and diversity of synthesized data.**
> Although synthetic data have the potential to be directly used to solve a downstream task without relying on pre-trained models (i.e. CLIP), they only deliver inferior performance as analyzed in Sec 3.1. The highest performance on the CIFAR-100 classification task is 28.74%, which is much lower than the zero-shot performance of CLIP (35.35%). This might be attributed to the quality and diversity of real-world data. The pre-trained CLIP model benefits from diverse real-world data which significantly improves the generalization ability of the pre-trained model. In our current exploration, we also analyzed the diversity and quality trade-off of synthetic data. When we increase the diversity of synthesized data, the quality of synthesized data might be sacrificed and noise will be increased (**see Table 2 “LE” in our paper, and Figure A.2 in appendix**)  which leads to performance degradation. We expect that further investigations on improving synthesis quality and diversities will bring new opportunities in this area which will be our future work.

---

### Author Response · Authors · 2022-11-18
**Fully Revised Draft**

Dear Reviewers and ACs:

Thank you very much for your constructive comments and insightful reviews which help improve our previous manuscript. We have carefully taken all the suggestions and made major changes to our previous draft, with the main changes marked blue in the draft.

Specifically, we have made the following changes:

**In the main paper:**
1. Added clarification of Classifier Tuning.
2. Added results of CLIP-ViT-B/16 backbone for zero-shot settings.
3. Added analysis of why synthetic data may harm pre-trained image encoder.
4. Added analysis for inferior performance in the training-from-scratch (CIFAR100, zero-shot) setting.
5. Added analysis of the diminished positive impacts of synthetic data when mixed with real data.
6. Added experiments of ViT-based backbone in the pre-training settings.
7. Fixed typos and revised language.

**In the supplementary material:**
1. Added analysis of different performance gains on different zero-shot tasks.
2. Added experiments of real-world data with domain shifts to study why synthetic data may harm pre-trained image encoder.
3. Added study of synthetic image number in zero/few-shot settings.
4. Added clarification of zero-shot performance differences.
5. Added FID and classification accuracy with a pre-trained classifier for synthetic data measurement.
6. Added visual examples of B, LE, LE+CF generated synthetic data.
7. Added visual examples of synthetic data for different datasets.
8. Added explanation for the selected 17 datasets in the zero-shot settings.
9. Added details of phase-wise training and mix training.

Thank you very much again!

Best regards, Paper 220 Authors

---

### Author Response · Authors · 2022-11-23
**Author Feedback for All Reviewers**

Dear Reviewers and ACs:

Thank you so much for your time and efforts in assessing our paper. Hope our rebuttal has addressed your concerns. We are happy to discuss with you further if you still have other concerns. Thanks for helping improve our paper.

Best regards, Paper 220 Authors

---

### Decision · Program_Chairs · 2023-01-20

**Decision:**

Accept: notable-top-25%

**Justification For Why Not Higher Score:**

All reviewers are in favor of accepting this paper, finding the experiments to be thorough and the results strong. I agree; I think this is very timely work that will be of high interest to the community, hence I suggest a spotlight presentation. I do not think the paper quite reaches the level of oral as there is a fair amount of prior work on using generative models for synthetic training data, and the methods in the current paper may be somewhat expected given this prior work plus the popularity of text-to-image models.

**Justification For Why Not Lower Score:**

See above.

**Metareview: Summary, Strengths And Weaknesses:**

Summary:
This paper uses a text-to-image model to generate synthetic training data for vision systems, with applications to zero-shot and few-shot learning, as well as pretraining. Labeled synthetic data is created by conditioning on text describing the class label. Experiments show that this approach boosts performance over using no synthetic data, and in the pre-training regime can be competitive with pretraining on real data.

Strengths:
* Timely investigation
* Extensive experiments
* Promising results in several settings

Weaknesses:
* High time complexity relative to modest performance gains



**Note From Pc:**

if the above contains the word "oral" or "spotlight" please see: "oral" presentation means -> notable-top-5% and "spotlight" means -> notable-top-25%. As stated in our emails, we are disassociating presentation type from AC recommendations